# Deciphering miRNAs’ Action through miRNA Editing

**DOI:** 10.3390/ijms20246249

**Published:** 2019-12-11

**Authors:** Marta Correia de Sousa, Monika Gjorgjieva, Dobrochna Dolicka, Cyril Sobolewski, Michelangelo Foti

**Affiliations:** Department of Cell Physiology and Metabolism, Faculty of Medicine, University of Geneva, CH-1211 Geneva, Switzerland; Marta.Sousa@unige.ch (M.C.d.S.); Monika.Gjorgjieva@unige.ch (M.G.); Dobrochna.Dolicka@unige.ch (D.D.); Cyril.Sobolewski@unige.ch (C.S.)

**Keywords:** microRNA regulation, miRNA editing, ADARs, APOBECs

## Abstract

MicroRNAs (miRNAs) are small non-coding RNAs with the capability of modulating gene expression at the post-transcriptional level either by inhibiting messenger RNA (mRNA) translation or by promoting mRNA degradation. The outcome of a myriad of physiological processes and pathologies, including cancer, cardiovascular and metabolic diseases, relies highly on miRNAs. However, deciphering the precise roles of specific miRNAs in these pathophysiological contexts is challenging due to the high levels of complexity of their actions. Indeed, regulation of mRNA expression by miRNAs is frequently cell/organ specific; highly dependent on the stress and metabolic status of the organism; and often poorly correlated with miRNA expression levels. Such biological features of miRNAs suggest that various regulatory mechanisms control not only their expression, but also their activity and/or bioavailability. Several mechanisms have been described to modulate miRNA action, including genetic polymorphisms, methylation of miRNA promoters, asymmetric miRNA strand selection, interactions with RNA-binding proteins (RBPs) or other coding/non-coding RNAs. Moreover, nucleotide modifications (A-to-I or C-to-U) within the miRNA sequences at different stages of their maturation are also critical for their functionality. This regulatory mechanism called “RNA editing” involves specific enzymes of the adenosine/cytidine deaminase family, which trigger single nucleotide changes in primary miRNAs. These nucleotide modifications greatly influence a miRNA’s stability, maturation and activity by changing its specificity towards target mRNAs. Understanding how editing events impact miRNA’s ability to regulate stress responses in cells and organs, or the development of specific pathologies, e.g., metabolic diseases or cancer, should not only deepen our knowledge of molecular mechanisms underlying complex diseases, but can also facilitate the design of new therapeutic approaches based on miRNA targeting. Herein, we will discuss the current knowledge on miRNA editing and how this mechanism regulates miRNA biogenesis and activity.

## 1. Introduction

MicroRNAs (miRNAs) are small non-coding RNAs (ncRNAs) highly conserved among species that modulate gene expression, mainly through translational inhibition or degradation of messenger RNAs (mRNA). However, the biology of miRNAs is much more complex than initially thought [1,2]. Indeed, recent advances in the field have uncovered numerous molecular mechanisms, which tightly regulate their biogenesis, maturation and action in a cell-dependent manner under physiological conditions or in diseases. For instance, single nucleotide polymorphisms; histone or DNA methylation; asymmetric miRNA strand selection; interactions with other coding and non-coding RNA molecules or RNA-binding proteins (RBPs); and RNA editing, are all recently identified mechanisms regulating miRNA biogenesis or activity. Among those new regulatory mechanisms, the relevance of miRNA editing for their functions is highly debated, but progressions in our understanding of these mechanisms are currently restricted by technological limitations specifically related to bioinformatic analyses of high-throughput RNA-seq approaches. Convincing evidence has been provided showing that a single nucleotide change on a primary miRNA molecule can greatly influence its stability and maturation, or can alter its activity by retargeting miRNAs towards other messenger RNAs (mRNAs), as shown in the context of several cancers or cellular stresses; e.g., hypoxia or endoplasmic reticulum (ER) stress [3,4].

This review discusses how miRNA editing affects their functions in physiological and pathological conditions, and currently available approaches to investigating these mechanisms.

## 2. The Complex World of miRNA Biology—From Biogenesis to Action

In the canonical biosynthetic pathway, a primary miRNA (pri-miRNA) is transcribed, recognized by a microprocessor complex, including the enzymes double-stranded RNA-specific endoribonuclease (DROSHA) and DiGeorge syndrome critical region gene 8 (DGCR8), and cleaved to form a precursor miRNA (pre-miRNA). Pre-miRNAs are then exported to the cytoplasm for further processing by the enzyme DICER and co-factors, such as the protein activator of protein kinase R (PACT) or the Trans-activation response RNA-binding protein (TRBP) [5]. The mature miRNA duplex is finally loaded into a multi-protein complex, the RNA-induced silencing complex (RISC) and one selected miRNA strand (-5p or -3p) binds to the Argonaute (AGO) protein, which guides the complex to its target mRNA [6]. However, miRNAs’ biogenesis and maturation appear to be much more complex and tightly regulated processes, under the control of multiple cellular factors sensitive to the physiopathological statuses of the cells and their environments. At the DNA level, single nucleotide polymorphisms (SNPs) and epigenetic control of transcription through classical mechanisms of acetylation/methylation of DNA/histones represent a first level of miRNA regulation governing their action (Figure 1a,b). Biosynthesis and maturation of miRNAs can also be influenced by RNA-binding proteins (RBPs), which can interact with key enzymes in these processes, such as DROSHA/DGCR8/DICER and the RISC complex (Figure 1c,d) [7]. Examples of such mechanisms are illustrated by the inhibition of let-7 miRNA processing induced by Lin28 binding [8,9,10] or stabilization of pri- and/or pre-miR-144 by BUD13 and Interleukin Enhancer Binding Factor 3 (ILF3) that leads to increased levels of mature forms [11]. Finally, editing of pri/pre-miRNAs is also an important mechanism modulating the biosynthesis and maturation of specific miRNAs, in addition to deeply impacting miRNAs’ actions on their targets (Figure 1c,d) [12,13,14].

How mature miRNAs regulate gene expression is, further, dependent on multiple factors that may vary with the cellular context and the cell environment [15]. The strand of the miRNA (-5p or -3p strand), which is degraded (passenger strand) or incorporated in the RISC complex (guide strand) determines the set of target mRNAs. The specificity of the RISC complex’s action towards mRNAs highly depends on the complementarity between the miRNA response elements (MRE) on the mRNA and the seed sequence on the miRNA strand. In addition, the degree of complementarity between MRE and seed sequences usually dictates whether the mRNA is degraded or whether its translation is blocked [16]. Of note, although most miRNAs interact with the 3′ untranslated regions (UTR) of target mRNAs to inhibit their expression, interactions of miRNAs with gene promoters, 5′UTRs or coding sequences have been described and may result in distinct outcomes. For example, activation of gene expression instead of repression might occur in specific conditions [1,17,18,19].

Another important feature of the miRNA biology is that a single miRNA can target hundreds of mRNAs, thereby regulating whole networks of proteins. Conversely, one mRNA can be targeted by several miRNAs [16,20]. There are two major consequences associated with these properties: (1) various miRNAs and other factors may compete for binding sites on a particular mRNA (e.g., on the 3′-UTR); (2) variations in the stoichiometry of targets for a specific miRNA, and localization of these targets within distinct cell compartments, can deeply affect the expected interaction between the miRNA and a particular mRNA. In this regard, miRNAs can also traffic between various intracellular compartments (e.g., nucleus, cytoplasm, stress granules and mitochondria) under stress conditions (e.g., starvation or hypoxia), thereby either modulating transcriptional/translational rates of target mRNAs within specific intracellular compartments or being secreted as paracrine factors mediating intercellular communication [4,17,21,22,23]. In addition, expression of other endogenous competing RNAs (ceRNAs), such as pseudogenes, circular RNAs and long ncRNAs (lncRNAs), can act as “sponges” and impair specific miRNA-mRNA interactions (Figure 1d) [20]. Finally, other cellular factors can interact with mature miRNAs and modulate their activity, as it has been demonstrated for AU-rich element-binding proteins (AUBPs) [16]. All these mechanisms suggest that expression and bioactivity of a specific miRNA within a particular cell type or tissue can be highly uncoupled and that observed alterations of miRNA expression do not necessarily mirror the activity of this molecule, potentially leading to misinterpretation of its relevance in pathophysiological conditions. In this regard, several tools have been developed; e.g., reporter gene constructs harboring multiple MREs for the miRNA of interest, to assess the bioavailability and activity of a specific miRNA in parallel to its expression level [24].

## 3. Editing of miRNA

RNA editing refers to specific modifications in an RNA molecule that result in alterations of the RNA sequence compared to the one encoded by the genome. RNA editing events occur in all eukaryotes, although the molecular mechanisms are taxon dependent and may involve different enzymes, leading to different RNA modifications, particularly between plants and animals [25,26]. For instance, substitutional RNA editing is a mechanism occurring across all metazoans, and this process is characterized by nucleotide modification mediated by deaminases of, principally, two distinct families. These include adenosine deaminases acting on RNA (ADARs) and cytidine deaminases from the AID/APOBEC protein family (activation induced cytidine deaminases/apolipoprotein B mRNA editing enzyme cytidine deaminases). ADARs are responsible for deamination of adenosine (A) to inosine (I), whereas AID/APOBEC deaminate cytidine (C) to uridine (U). These reactions are known as A-to-I editing and C-to-U editing, respectively [25].

Enzymes triggering RNA editing are essential for several processes requiring cell growth and adaptation (e.g., embryogenesis, proliferation, immunity and neural plasticity) [27,28,29]. In this respect, activity of editing enzymes can lead to various functional cellular outcomes depending on the RNA molecules being modified and of the type of modification. Indeed, nucleotide substitution within RNA sequences may occur in distinct regions of the transcript and affect the final protein product differently: through disruption of the reading frame, alterations of splicing sites or modifications of regions essential for RNA interference mechanisms [30]. Although the consequences of editing on coding RNA are evident, the frequency of RNA editing events is higher in non-coding sequences of the transcriptome [31,32]. The identification of identical A-to-I miRNA editing events (on miR-140, miR-301a and miR-455) in both mammals and birds, further suggests that miRNA editing is a highly conserved mechanism, which appears to occur frequently within seed sequences of miRNAs [33]. In light of our current knowledge, it is clear that editing of non-coding RNA sequences, such as miRNAs, might impair their regulatory functions, thereby affecting, directly or indirectly, cellular processes and homeostasis [13].

### 3.1. ADAR

ADAR proteins catalyze deamination of A-to-I on mRNAs. This protein family is highly conserved in metazoans [34], but the number of genes and isoforms are species-specific. In mammals, three *ADAR* genes were identified—*ADAR1*, *ADAR2* and *ADAR3*—but only the first two seem to encode proteins with catalytic activity. *ADAR1* is ubiquitously expressed to different extents depending on the tissue or cell type. Regarding *ADAR2*, while the GEPIA2 database indicates a lower but also ubiquitous distribution of *ADAR1*, the enzyme was reported to be highly expressed in the brain, but at lower levels in other tissues (e.g., lung, kidney, testis and vascular tissues). Finally, *ADAR3* expression was shown to be mostly brain and testis-specific (Figure 2a,b) [34,35,36,37]. *ADAR1* gene encodes two distinct isoforms that localize in different cell compartments—the full-length ADAR1p150 is inducible by interferon and mostly located in the cytoplasm, but can translocate also into the nucleus [34,35]. The N-terminally truncated ADAR1p110 (constitutively expressed) and ADAR2 are exclusively located in the nucleus [38].

ADARs are required during embryogenesis of mammals, since knock-out models for both ADAR1 and ADAR2 are not viable and die during embryonic development or shortly after birth due to over-stimulation of the immune system; hepatic and hematopoietic disorders (ADAR1 knockout); or from neurological disorders (ADAR2 knockout) [38,39,40,41]. Despite these striking phenotypes, very little is known about the specificity of ADAR editing activity, which is still debated [42]. The secondary structure of dsRNA appears to impact the editing frequency. For example, perfect stretches on dsRNA generally represent hyper-edited sites independently of their sequence. In contrast, RNA loops require specific nucleotide pairings in order to be edited [38]. In addition, editing activity of ADAR enzymes seems to be affected by other factors independent of the RNA structure [42].

Based on their secondary structure, A-to-I editing could occur in pri-, pre- and mature miRNAs, although the probability of editing events is likely dependent on the lengths of these different molecules [38,42,43] (Figure 3a). RNA cis-elements and the recruited ADAR isoform might also be important for miRNA editing, as highlighted by several studies, suggesting a preferred editing activity of nuclear ADARs, i.e., ADAR2 and ADAR1p110, depending on miRNA sequence characteristics [38]. This concept is further refined by data suggesting that nucleic forms of ADARs are responsible for editing of pri- and pre-miRNA, while cytosolic forms are responsible for editing of mature miRNAs before or during AGO2 incorporation [30]. However, shuttling of ADAR1p150 and specific mature miRNAs, through mechanisms involving members of the GW-182 protein family, between the nucleus and the cytoplasm indicates that A-to-I editing is not as compartmentalized as suggested [30,44].

As discussed later, editing by ADARs can greatly impact miRNA biogenesis and function (Figure 3a). However, recent studies also revealed that ADARs can promote miRNA biogenesis, independently of its editing activity, through direct interactions with the DICER enzyme [45].

### 3.2. AID/APOBEC

The AID/APOBEC family includes several enzymes sharing a similar structure but differing in terms of function, with only few of them, i.e., APOBEC1, APOBEC3A and APOBEC3G, having C-to-U RNA editing activities [29,46,47,48]. Similarly to ADAR proteins, APOBECs have different intracellular distributions. Both APOBEC1 and APOBEC3A shuttle from the nucleus to the cytoplasm and vice versa [47], while APOBEC3G is restrained to the cytoplasm under physiological conditions [49].

In contrast to *APOBEC1*, *APOBEC3A* and *3G* are expressed in most tissues (Figure 2a), but APOBEC1 is the most well characterized isoform of the AID/APOBEC family [50]. Loss of APOBEC1 does not alter embryonic development but it impairs lipoprotein metabolism by editing apolipoprotein B (apoB) RNA and it affects neurological function [48,49,50,51]. In contrast to information collected from DICE database (Figure 2b), the APOBEC1 deaminase was reported to be significantly expressed in immune cells, where it exerts RNA editing activity on 3′-UTRs of numerous mRNAs [48,50,52]. Regarding APOBEC3A and APOBEC3G, very little is known about their functional roles. Recently, APOBEC3A was shown to be expressed mainly in immune cells, specifically in monocytes and macrophages, and to be upregulated by interferon type 1 (IFN-1) [52]. On the other hand, APOBEC3G binds DNA and induces genotoxicity when not restricted to the cytoplasm [49]. Interestingly, however, APOBEC3G was suggested to counteract the miRNA-mediated inhibition of gene expression independently of its potential editing activity. Indeed, APOBEC3G might control miRNAs’ actions by interfering with the RISC complex assembly (Figure 3b), a mechanism further supported by the reported interaction of APOBEC3G with Moloney leukemia virus 10 protein (MOV10), an essential RISC complex component [53,54].

Finally, APOBEC-mediated editing occurs preferentially on single-stranded RNA (ssRNA) molecules with a looped conformation and in AU-rich sequences [48,52]. Oligomerization of APOBEC proteins might be also required for their editing activities, and interactions with other factors [47]. In this regard, RNA-binding motif-47 (RMB47) was shown to modulate APOBEC1 editing activity by recruiting the deaminase to specific transcripts [48].

Occurrence of C-to-U editing in miRNA and its functional consequences on miRNA activity or specificity are still unclear since very few studies are available, compared to investigations of A-to-I substitution in miRNA [32,55]. It is likely, however, that APOBEC’s editing of 3′UTRs of mRNAs may significantly modify MREs, and therefore, change miRNAs’ specificities for particular transcripts (Figure 3b) [50,56].

## 4. Editing in miRNA Biogenesis and Activity

### 4.1. miRNA Maturation

Several lines of evidence support a key role for RNA editing on the maturation processes of miRNAs (Figure 1c). First, ablation of ADARs in *Caenorhabditis elegans* and in mice was reported to compromise pri- or pre-miRNA processing, resulting in altered miRNA levels [57]. Second, in human cell lines (HEK293 cells), destabilization of miRNA structure by editing events on regions outside of seed sequences resulted in alterations of (i) miRNA processing by the DROSHA/DICER complex (e.g., miR-151, let7-g, miR-33, miR-133a2, miR-197, miR-203 and miR-379) [58,59], and (ii) miRNA strand-selection and loading onto AGO [60,61]. In addition, adenosine deamination was shown to introduce specific mismatches potentially recognized by inosine-specific RBPs with endonuclease activity such as Tudor staphylococcal nuclease (Tudor-SN) and endonuclease V (ENDOV) [62,63,64]. Such events affect both level of mature miRNAs and their activities (Figure 3a). It was, for example, shown that editing of pri-miR-142 in HEK293 cells inhibited DROSHA cleavage and led to Tudor-SN degradation of the edited miRNA [12]. Consistently with these findings, increased levels of miR-142 were found in hematopoietic tissues of B-lineage-specific ADAR1 knock-out mice compared to wild-type controls [12]. Furthermore, editing of pri- or pre-miR-151 was reported to inhibit DICER processing [59], while another recent report indicates that ADAR2 edits precursors of miR-221, miR-222 and miR-21 in the brain, thereby impairing processing and maturation of these oncogenic miRNAs [45].

Conversely, editing can also favor the stability, processing and maturation of miRNAs (Figure 3a). For example, editing of primary miR-376a-1 at position +4 increases the stability of the molecule compared to the unedited one. This specific editing event was shown to be conserved between different human tissues (brain and placenta) and in primate brain tissues [65]. Miscellaneous effects of miRNA editing were also observed in gastric cancer cells (AGS and MKN 45 cell lines), where edited miRNAs are either upregulated, e.g., miR-345-5p, miR-149-5p, let-7a-5p and miR-221-5p, or downregulated, e.g., miR-146b-5p, miR-148a-3p miR-22-3p and miR-302a-3p, following ADAR1p150 knock down [66]. Finally, editing regulates the expression of polycistronic miRNA clusters, as suggested by experimental evidence in *Drosophila melanogaster*, where ADAR editing of primary let-7 cluster at different sites selectively modify the processing of distinct miRNAs depending on the editing position [67].

### 4.2. miRNA–mRNA Interactions

Editing of seed sequences of miRNAs may not only impair their target recognition but also redirect them to a different network of mRNAs or competing RNAs (Figure 3) [13,16]. This is typically illustrated by the case of the edited and unedited forms of miR-455-5p, which recognize different target genes, thereby contributing to distinct outcomes in melanoma development/progression [68]. Another striking example of this effect is mir-376a*, which targets different mRNAs under its edited and unedited form, i.e., autocrine motility factor receptor (AMFR) and Ras-related protein 2A (RAP2A) respectively, thereby promoting glioblastoma development and progression when a decreased editing capacity occurs in the brain [69]. miRNA editing (e.g., A-to-I substitutions) may have also other functional consequences as in the case of miR-376, miR-22 and miR-191, where A to-I substitutions impair the thermodynamics of nucleotide pairing, leading to weaker miRNA MRE binding of known targets of the unedited forms, consequently, modifying the silencing efficacy of these miRNAs [70]. Of note, editing activity on miRNAs is limited by the stoichiometry of editing enzymes expressed in cells; thus, both edited and unedited forms of miRNAs are usually found in the same cell and in variable ratios depending on the cell type. mRNA targets modulated by a single miRNA are, therefore, highly dependent of this cellular edited/unedited ratio, as well as the related cellular processes.

RNA editing can also modulate miRNA activity indirectly by modifying MREs of their corresponding target mRNAs (Figure 3c) [71,72,73]. For example, loss of edited sites in the 3′-UTR of phosphatase and actin regulator 4 gene (*PHACTR4*) due to downregulation of ADAR1 prevented the binding of miR-196a-3p, resulting in higher protein levels of PHACTR4 [66]. As well, ADAR1 knockdown in the hepatic cell line Huh-7 correlated with upregulation of human aryl hydrocarbon receptor (AhR) due to loss of miR-378 target site on its 3′-UTR [74]. Finally, *APOBEC1* knockout in mice led to 238 C-to-U substitutions on the 3′-UTRs of several genes, modifying the pattern of miRNAs susceptible to modulate their expressions [56].

### 4.3. miRNA–ceRNA Interactions

Competing endogenous RNAs, including pseudogenes, circular RNAs and lncRNAs, play a major role in the regulation of miRNA activity by acting as natural miRNA sponges or decoys [75]. Indeed, multiple MREs are present in ceRNAs and enable direct interactions with miRNAs, thereby preventing binding of these miRNAs to their respective target mRNAs (Figure 1d). The role of ceRNAs in various cellular processes (e.g., cell cycle, apoptosis) is well established, and alteration of their expression contributes to the onset of various pathologies [76,77,78]. One representative and well-studied example of such regulation is provided by the phosphatase and tensin homologue, pseudogene 1 (*PTENP1)*, which can sponge various miRNAs (e.g., miR-21, miR-106b and miR-93) targeting the 3′UTR of the tumor suppressor gene phosphatase and tensin homologue (*PTEN*) [79,80]. Accordingly, *PTENP1* acts as a tumor suppressor, and its loss leads to a reduction of PTEN expression in various cancers [79,80,81,82]. It is, therefore, clear that editing on miRNA seed-sequences or their corresponding MREs on ceRNAs may impair their interaction, leading to imbalance in these regulatory mechanisms and functional outcomes. Of note, emerging evidence indicates that lncRNAs are also subjected to RNA editing [83,84]. The potential impact of editing on ceRNA–miRNA interactions is currently likely underestimated, but it may represent an important mechanism of epigenetic plasticity.

### 4.4. Regulation of the miRNA Editing Machinery by RNA-Binding Proteins (RBPs)

RNA-binding proteins are a family of multi-functional proteins capable to directly interact with mature transcripts and miRNAs, or to form complexes with other regulatory factors, thereby modulating miRNA editing, processing or activity (Figure 1c,d) [85,86,87]. RBPs binding to miRNAs or competition of RBPs and miRNAs for specific binding sites on mRNA targets are also important regulatory mechanisms of gene expression [85,86] that can potentially be deeply affected by RNA editing. For example, this is the case of human antigen R (HuR), an RBP with specificity for AU-rich elements, which competes with miRNAs (e.g., miR-21) for binding sites in the 3′UTR of target mRNAs (i.e., PDCD4, proinflammatory tumor suppressor protein programmed cell death 4), thereby stabilizing this transcript as opposed to the effects of miRNAs. Binding of HuR to these specific 3′UTR sequences can be modulated by A-to-I editing of target mRNAs [85]. In addition, binding sites for HuR are present in close proximity to ADAR1 binding sites, and both enzymes appear to directly or indirectly interact to regulate transcript stability [88,89]. For example, ADAR1 and ADAR2 prevent destabilization of the Cat2 transcribed nuclear RNA (CTN) transcript, which is mediated by HuR in complex with poly(A)-specific ribonuclease deadenylase (PARN), by competing for specific binding sites within the target transcript [90,91].

Finally, RBP––miRNA interactions can also be regulated by editing proteins, as illustrated by APOBEC3G, which counteracts the inhibitory effect of dead-end protein homolog 1 (DND1) on miRNA–mRNA interactions, thereby restoring for example the inhibitory activities of miR-372 and miR-206 on their mRNA targets [87].

## 5. Tools to Study miRNA Editing

Classical genetic approaches of gain and/or loss of function of editing enzymes in rodents have been performed to investigate the biological relevance of RNA editing mechanisms [92]. However, knockout of genes such as *Adar* leads to non-viable embryos; thus, preventing further analyses [92]. Therefore, most of the information regarding the biological role of these enzymes has been gained from in vitro experiments designed to address the impact of editing activity on miRNAs, as described in other sections. In this regard, methodologies to modify nucleic acids within living cells have been developed; e.g., site-directed RNA editing (SRDE) systems. SRDE is based on cell transfection of chimeric proteins, including deaminase domains (DD) from ADAR1/2 and guide RNAs molecules allowing targeting of the chimera to the RNA to edit [93,94,95]. SRDE techniques were developed using a bacteriophage λn peptide fused to ADAR2 DD. The λn peptide recognizes a specific boxB hairpin RNA, which, in addition to the sequence complementary to the desired target mRNA, is encoded by the guide RNA [95,96]. The methodology evolved with the generation of either wild-type or mutated ADAR1/2 DD fused to a SNAP-tag, which has the advantage of recognizing and binding to chemically modified guide RNAs [94,97]. More recently, the CRISPR-Cas13b RNA editing for programmable A-to-I replacement (REPAIR) system has been established by fusing a Cas13b protein and wild-type or mutated ADAR1/2 DD, and it acts similarly to the CRISPR-Cas9 system [93]. This REPAIR system also triggers RNA editing under the control of guide RNAs. These methodologies allow, to some extent, guided A-to-I substitution in the transcriptome; therefore, representing key tools with which to decipher the functional consequences of miRNA editing. Optimization of these methods aims at overcoming significant issues associated with these approaches, such as off-target effects, generation of complex guide RNAs molecules and variability in the editing efficiency of large RNA targets in particular. In addition, since SRDE is currently not used to edit miRNAs, whether this approach is applicable to small size RNAs remains to be evaluated. Finally, a powerful alternative to investigate the functional relevance of miRNA editing remains to be the usage of synthetic modified oligonucleotides mimicking or inhibiting the edited miRNA of interest, one which can efficiently be incorporated in vitro by cultured cells or in vivo following venous injections [20,70].

Regarding the detection of RNA editing events per se, all currently available methods are based on comparison of complementary DNA (cDNA) sequences obtained from reverse transcribed (RT) RNAs with the original genomic information of samples under consideration or with miRNA sequences provided by miRNA databases [98]. The identification of A-to-G or C-to-T substitutions (since during reverse transcription I is recognized as G and U as T) delineates edited sites, provided that sequencing techniques are accurate and do not introduce errors. Despite the simplistic concept, identification and validation of these modifications specifically in miRNAs, remains complex due in particular, to differences between methodological protocols (e.g., between commercially available kits for RNA isolation) used to isolate and to prepare libraries of miRNAs. Indeed, distinct biases are associated with different methodological protocols and their efficiencies to prepare miRNA libraries, thereby leading to potential discrepant analyses (e.g., amounts of false positives) between studies [99]. To identify editing events in small regions of interest, e.g., precursor/mature miRNAs or intronic/untranslated regions of genes harboring MREs, Sanger sequencing on gDNA and cDNA amplified by PCR usually allows one to recognize edited sites. However, interpretation of the data might be distorted due to the fact that not all transcripts of the same sequence might be modified [100]. Conversely, to detect multiple editing events in the same transcript or sample, next generation sequencing (NGS) techniques are more appropriate [101].

Several bioinformatics tools and pipelines have been designed to eliminate potential artefacts and false positive editing events when analyzing sequencing data that typically follow similar workflows. In addition, processing pipelines can be differentially designed to take into consideration various parameters used to filter and annotate reads as miRNAs; i.e., size exclusion [102], minimum free energy to predict secondary structure, number and type of mismatches and alignments with either genomic libraries [103] or available miRNA databases (e.g., miRbase or mirGeneDB) [102,104]. Processing pipelines may also vary in their depths of analyses, some of them assessing only the abundance of edited miRNA forms (e.g., miRSeqNovel [105], isomiReX [106], IsomiRage [107], MIRPIPE [108], miRge 2.0 [102] or mirPRo [109]), while others allow one to identify targets of edited miRNAs (e.g., SeqBuster [110], iMir [111] and Prost! [112]) and related genes and pathway enrichment, and protein–protein interactions (e.g., CPSS 2.0 [113], miRGator v3.0 [114], miRTools 2.0 [115] and DeAnniso [116]). Finally, whether available algorithms are user-friendly or not, free or paid and allow parameters changes or not, often depends of whether these tools are available online (e.g., CPSS2.0 [113]; miRGator v3.0 [114]; miRTools 2.0 [115]; MIRPIPE [108]; sRNAbench [117]; DeAnnIso [116]), or require a local installation using the terminal, or are provided as independent packages for use with Python/R (e.g., SeqBuster [110], miRSeqNovel [105], iMir [111], IsomiRage [107], miRge 2.0 [102], mirPRo [109] and Prost! [112]). The more recent online algorithms are usually faster, but do not always allow one to analyze data in batch mode [108,117]. This variability of protocols and pipelines dedicated to the identification of miRNA editing sites from sequencing data can of course strengthen the reliability of data obtained through different bioinformatic approaches, but also can lead to differences and inconsistencies between studies.

Finally, recent progresses of single-cell (sc) omics-related methods should help with investigating the variability of miRNA profiles and editing between single cells within the same tissue. In this regard, a recent report by Wang et al. (2019) highlighted different miRNA and mRNA profiles between single cells expanded from a K562 cell line, as well as potential molecular mechanisms underlying the observed transcriptomic variability [118]. Improvements in the methods to prepare and process transcript libraries coupled to scRNA-seq techniques should greatly advance, in the future, our understanding of the functional role of miRNA editing in transcriptome variability of single cells [119].

## 6. miRNA-Dependent Regulation of Editing—Closing the Loop

Regulation of ADAR and APOBEC RNA editing enzymes is currently poorly understood, but based on bioinformatic predictions (Figure 4) and nascent experimental evidence, it is likely that reciprocal regulation between miRNA and editing enzymes may occur. In this regard, miRwalk 2.0 database predicts several miRNAs to target *ADAR1*, *ADAR2* and *ADAR3* (Figure 4a). However, the functional relevance of miRNA-based regulation of editing enzymes remains to be firmly established. Currently, only one study has reported downregulation of ADAR1 by miR-17 and miR-432 in melanoma cells, a mechanism suggested to foster tumor growth [120]. Although nothing is known about APOBECs’ regulation by miRNAs, an interesting observation supports a functional regulation by miRNAs of this class of enzymes. Indeed, genetic polymorphisms exist where the *APOBEC3A* coding sequence is fused to the *APOBEC3B* 3′UTR. This hybrid fusion product is overexpressed compared to wild type APOBEC3A, and hyper editing activity on nuclear DNA is observed [121]. One hypothesis supporting this phenotype is that *APOBEC3A* expression is repressed to some extent through 3′UTR-dependent mechanisms, potentially through the action of miRNAs, and this repression is abolished when its 3′UTR sequence is exchanged with the one of *APOBEC3B*. Consistent with this concept, our bioinformatic analysis of the *APOBEC3A* 3′UTR using the miRWalk 2.0 database reveals the presence of several potential miRNA-binding sites (Figure 4b). Further analyses and experimental evidence are now required to assess the functional relevance of reciprocal regulatory interactions between miRNAs and editing enzymes in both physiological and pathological conditions.

## 7. miRNA Editing in Pathophysiological Processes

### 7.1. Development

In early stages of human embryogenesis (eight cells to the morula stage), RNA editing activity was reported to be low [27], but whether this observation can be extrapolated to miRNA editing is unknown. Nevertheless, in murine E15 embryonic stage miRNA editing events were reported to be rare, but to increase with postnatal development [122]. An increased miRNA editing during postnatal development and aging was observed in humans, and macaques as well, and several miRNAs, such as miR-376b, miR-376c, miR-381, miR-379, miR-411 and miR-497, showed an age-correlated increase in editing frequency in both species [33]. A higher number of mature miRNAs was found in embryonic tissues from ADAR2-deficient and ADAR1/ADAR2-deficient mice, indicating that miRNA editing leads more to the inhibition of miRNA processing, rather than its enhancement [104]. Interestingly, in the same study, only slight differences in miRNA abundances were detected in embryos from ADAR2-deficient and ADAR1/ADAR2-deficient mice, emphasizing the relevance of ADAR2 for miRNA editing.

In both humans and mice, neural tissues display significantly higher editing levels than non-neural tissues [33]. An increased A-to-I editing in the mammalian brain throughout development was reported to lead to alterations of miR-381 and miR-376b, thereby affecting their target specificity, among which pumilio RNA binding family member 2 (PUM2), a translational repressor negatively regulating dendritic outgrowth, was identified [122]. Moreover, alterations of miRNA abundance by inhibiting ADAR activity resulted in synapsin 2 (SYN2) downregulation and upregulation of miR-153, miR-30 and miR-32, all predicted to target *SYN2* [104]. The impact of miRNA editing on the expression of SYN2, which contributes to synaptogenesis, and PUM2, which regulates dendritic outgrowth, are representative example illustrating the functional relevance miRNA editing in developmental processes [104,122].

### 7.2. Obesity and Metabolic Diseases

The relevance of miRNA editing in metabolic diseases has not been currently evaluated, but miRNAs are playing key roles in these disorders (see, for example, [20]) and several lines of evidence indicate that RNA editing enzymes might represent key regulators of metabolic processes [20]. For instance, ADAR1 and ADAR2 are upregulated and RNA editing is increased in β-cells of the pancreases of diet-induced obese and insulin-resistant mice [123]. In the same cells, *ADAR2* expression was further reported to be regulated by glucose and the nutritional status of mice, suggesting that RNA editing might regulate beta-cells’ activity and glucose metabolism [124]. In another study, transgenic mice overexpressing a catalytically inactive ADAR2 isoform were reported to develop hyperphagia-mediated obesity [125]. Finally, a decreased activity of APOBEC1 was shown to promote the unedited form of the ApoB protein, a phenotype associated with atherosclerosis [25,56]. As already mentioned, whether metabolic disorders related to deregulations of editing enzymes are mediated through miRNAs-dependent mechanisms remains unknown and needs to be evaluated [125]. 

### 7.3. Inflammation and Immunity

Although direct evidence supporting a role for miRNA editing in inflammation is scarce, RNA interference and editing, and dysregulations of editing enzymes, have been associated to the development and progression of several inflammatory conditions [103,104,105,106]. Based on DICE project database (https://dice-database.org/) analysis [126], ADAR and APOBEC editing enzymes are differentially expressed between immune cells subtypes, e.g., naïve and activated B and T cells, suggesting different editing activities associated with immune cell activation (Figure 2B). JAK2 signaling induced upregulation of ADAR1 expression in immune cells was experimentally confirmed and suggested to impact miRNA processing, since ADAR1 editing activity affected let-7 biogenesis in isolated hematopoietic progenitor cells and in the leukemic cell line K562 [127,128]. APOBEC3A was also reported to induce RNA editing in monocytes and macrophages following pro-inflammatory stimuli (i.e., hypoxia and interferon) leading to editing and downregulation of several genes, possibly via miRNA targeting [29]. Along the same line, increased editing of 3′-UTR mRNAs by APOBEC3G was observed in natural killer (NK) cells following hypoxic stimuli [129]. Finally, several miRNAs known to be significantly edited, e.g., miR-155 and miR-222, are essential for hematopoiesis and myeloid/lymphoid lineage commitment [130], and deeply so, for the functions and adaptability of immune cells [131,132].

The importance of miRNA editing in inflammatory processes does not only rely on the direct modulation of immune cells functions specifically. Indeed, miRNA editing was equally shown to contribute to changes in specific miRNA targets induced by hypoxia [3], a condition modulating immune responses and progression of related pathologic conditions (e.g., inflammatory bowel disease, liver diseases and cancers) [133]. Supporting this concept, 31 A-to-I editing events were observed in miRNAs of a human breast cancer cell line exposed to hypoxic conditions (e.g., miR-200b-3p, miR-148b, miR-27a-5p and -3p miR-421, etc.) with 83% of nucleotides substitution occurring in miRNA seed sequences [3].

### 7.4. Cancer

Similarly to metabolic diseases, the functional relevance of miRNA editing in cancer is still a field of research in its infancy. Most of the evidence suggesting a role for miRNA editing in cancer initiation and/or development results from observed alterations of the expression of editing enzymes in cancerous cells and in cancers of the nervous system, where editing activity is high. Editing enzymes display highly versatile expression levels, depending on the cancer type, when compared to normal tissue (Figure 5) [134]. *ADAR1*, which is the most expressed enzyme of the ADAR family, was found downregulated in adrenocortical carcinoma (ACC), while it was upregulated in cholangiocarcinoma (CHOL) [134]. In glioblastoma (GBM) both *ADAR2* and *ADAR3* expression are decreased, suggesting low levels of RNA editing [134]. Inhibition of A-to-I miRNA editing was further confirmed in high-grade gliomas to lead to an increase of unedited miRNAs, such as miR-376a [69]. Similarly, low editing of miR-589 in glioma was reported to change its specificity from disintegrin and metalloproteinase domain-containing protein 12 (ADAM12), a primary target of miR-589, towards protocadherin 9 (PCDH9), thereby promoting cell migration and invasion [135]. This example highlights the high potential of miRNA editing to influence carcinogenesis in specific organs.

Even though miRNA cytosine deamination by APOBECs remains controversial, expression of these enzymes (APOBEC1, 3A and 3G) is also altered in cancers (Figure 5). *APOBEC1* is upregulated in tumor samples from different adenocarcinomas; e.g., from the colon (COAD), the pancreas (PAAD), the rectum (READ) and the stomach (STAD). Other isoforms of APOBEC have been reported to have opposite deregulations in specific cancer types, such as in diffuse large B-cell lymphoma (DLBC), where APOBEC3A is downregulated while expression of the 3G isoform is increased (Figure 5). In contrast, both the 3A and 3G isoforms of APOBEC are overexpressed in acute myeloid leukemia (LAML) (Figure 5). A link between APOBEC3G and miRNA was previously suggested by Ding et al. (2012) [55], who observed that overexpression of APOBEC3G in colorectal liver metastasis, promoted cell migration and invasion due to the loss of miR-29-dependent repression of matrix metalloproteinase-2 (MMP-2). However, whether APOBEC3G-dependent miR-29 downregulation in this case relies on the APOBEC3G editing activity remains elusive [55]. Of interest, analyses of editing events occurring in different cancer types revealed that 3′ UTR regions were the most edited sites of several cancer-related transcripts. The authors of this study further confirmed that repression of a specific oncogene, i.e., mouse double minute 2 homolog (*MDM2*), by miR-200b/c was impaired due to extensive editing of the MDM2 3′UTR [73]. Finally, edits of specific miRNAs can also potentially be used as biomarkers or prognostic markers in cancer. In lung adenocarcinoma, several edited sites on mature miRNAs common to most of the tumoral samples were identified [103]. Deep analyses of data available from The Cancer Genome Atlas (TCGA), also allowed for some to correlate the prevalence of miRNA editing events with *ADAR1*/*2* expression, and to characterize A-to-I RNA editing hotspots in microRNAs of various cancers that correlate with tumor subtype and behavior [136].

Based on the above evidence, it is highly conceivable that alterations of the expression/activity of editing enzymes in cancer may represent key events contributing to carcinogenesis either by impacting directly miRNA sequences, or indirectly by modifying the 3′UTR sequences of miRNAs-regulated cellular factors driving carcinogenesis.

## 8. Conclusions

Since the discovery of miRNAs in 1993 [137], immense progress has been made in our knowledge of the regulation and role of these small non-coding RNAs in cell physiology and in a variety of diseases. The biology of miRNAs is much more complex than initially thought and new tools and approaches need to be developed in order to further unravel mechanisms regulating their actions. miRNA editing is a recently discovered mechanism that adds a layer of complexity in the coordinated action of miRNAs in a myriad of pathophysiological processes. An in-depth understanding of mechanisms regulating miRNA functions, such as miRNA editing, is a necessary prerequisite to not only comprehend the overall fundamentals of miRNA biology, but to envision therapeutic targeting of miRNAs in specific pathologies. Indeed, the degree and specificity of miRNA editing are important parameters to assess when considering, for example, the clinical use of synthetic nucleotides mimicking or inhibiting specific miRNAs, since they can impact therapeutic success. Furthermore, miRNA editing may provide us with important and specific biomarkers, particularly if edited miRNAs are secreted in the blood, for currently undetectable pathologies or for staging diseases such as cancers.

## Figures and Tables

**Figure 1 ijms-20-06249-f001:**
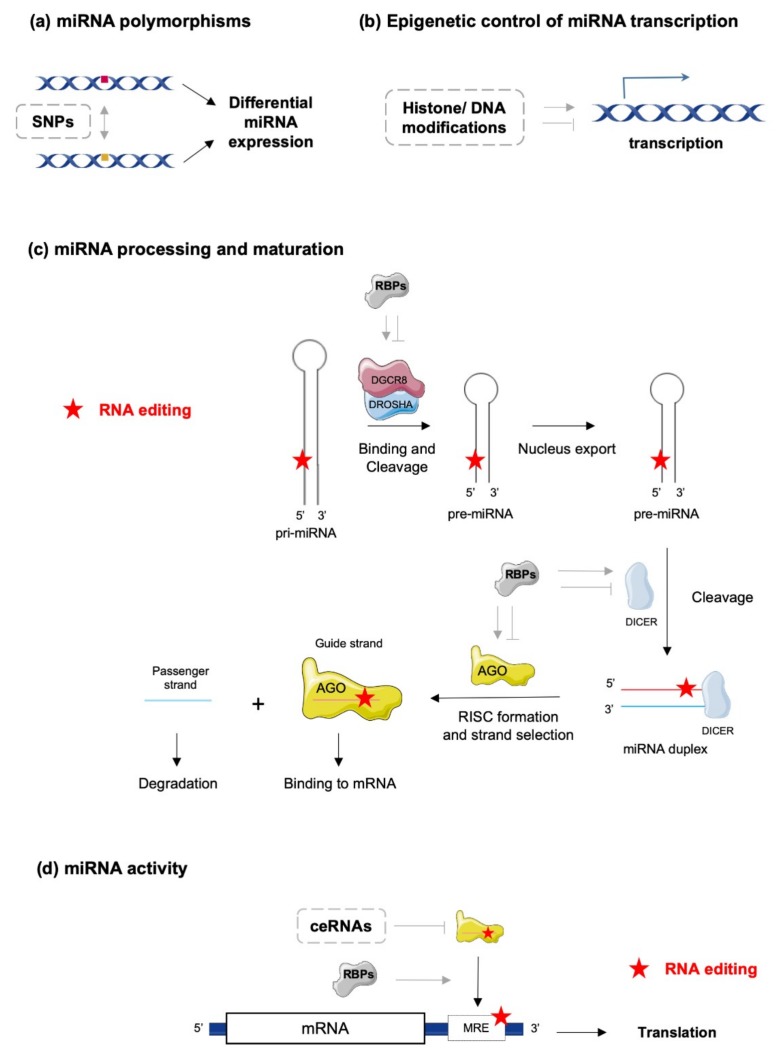
Molecular mechanisms controlling microRNA (miRNA) biogenesis and activity. (**a**) Single nucleotide polymorphisms (SNPs) in genomic regions encoding miRNAs can impact their processing and maturation, resulting in different expression levels between variants. (**b**) Epigenetic modifications such as histone acetylation or DNA methylation can modulate transcription efficiencies of miRNAs. (**c**) RNA editing (red stars) and RNA-binding proteins (RBPs) can interfere with miRNA processing and maturation by affecting DGCR8/DROSHA activity, nuclear export of pre-miRNAs, cleavage of pre-miRNA by DICER and incorporation of mature miRNA in the RISC (AGO) complex. (**d**) Mature miRNA activity can be regulated by alternative mechanisms including (i) competition with RBPs for the same binding site on a target mRNA, (ii) decoying of the miRNA–RISC complex by ceRNAs and (iii) RNA editing of miRNA response elements (MREs) on target mRNAs. Stimulatory and inhibitory effects are represented by pointed and blunt grey arrows, respectively.

**Figure 2 ijms-20-06249-f002:**
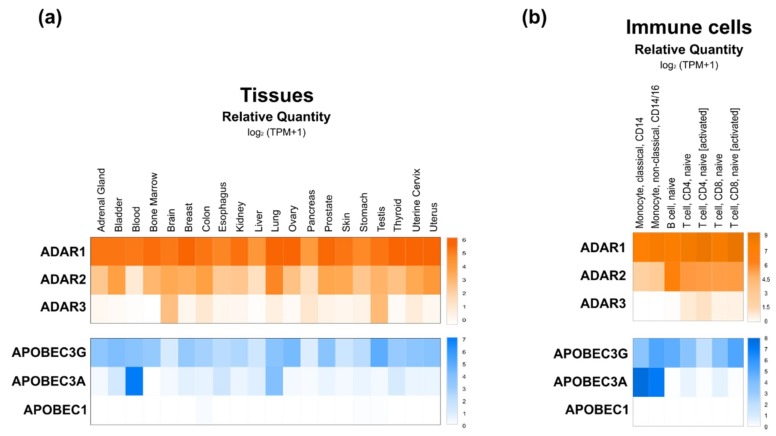
Expression patterns of *ADAR* and *APOBEC* editing enzymes. (**a**) Heat map showing mean relative expression (log2(transcripts per million (TPM) + 1)) of *ADARs* and *APOBECs* across different tissues. Expression levels of each gene were retrieved using GEPIA2 database (http://gepia2.cancer-pku.cn/#index), which include non-tumoral tissue samples from the Cancer Genome Atlas (TCGA) and normal tissue samples from the genotype-tissue expression (GTEx) projects. (**b**) Heat map showing mean relative expressions (log2 TPM + 1) of *ADAR*s and *APOBEC*s in subpopulations of immune cells, before or after activation. Expression levels of each gene were retrieved from DICE database (https://dice-database.org/). Both heat maps were designed using Morpheus (https://software.broadinstitute.org/morpheus).

**Figure 3 ijms-20-06249-f003:**
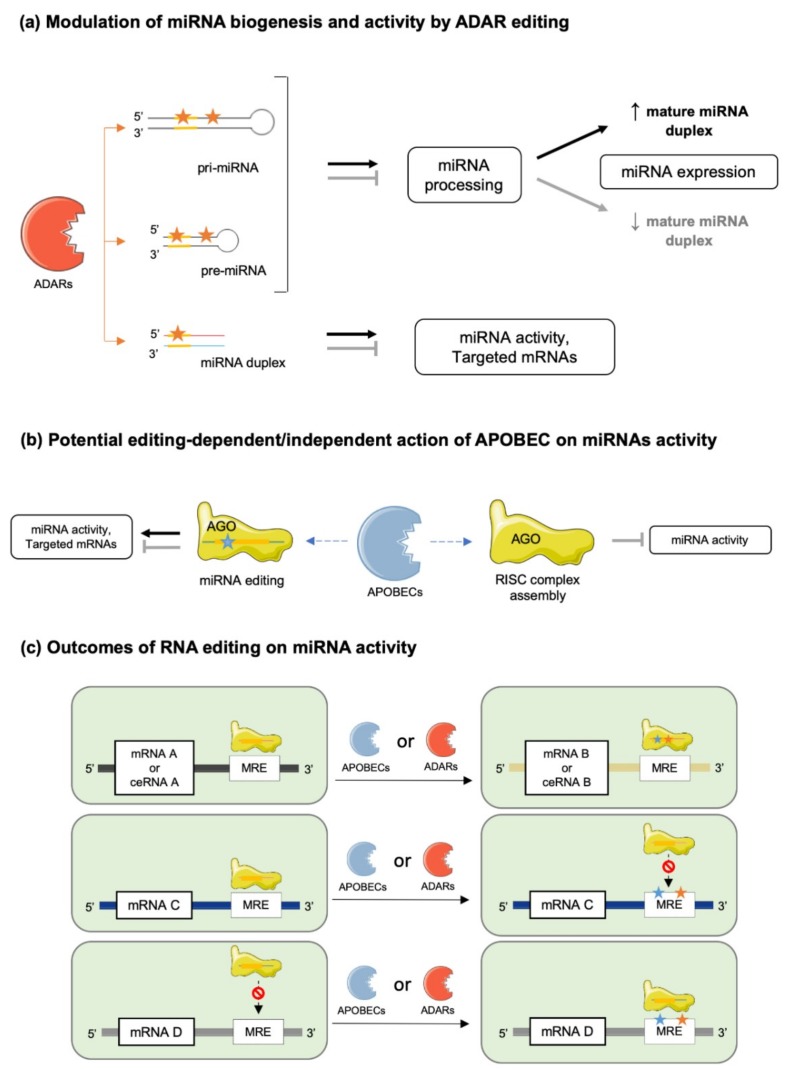
Functional outcomes of RNA editing on miRNA biogenesis and activity. (**a**) ADAR-mediated miRNA editing can occur in pri- and pre-miRNA, both within and outside of seed sequences, thereby affecting their processing (pointed black arrows: stimulation of processing; blunt grey arrows: inhibition of processing) and final expression levels of the mature miRNA duplexes. Editing can also occur on mature miRNA duplexes seed sequences either modulating their activities or changing their specificities to different target mRNAs. (**b**) Activity of APOBEC enzymes interfere with miRNA functions through editing-dependent or independent mechanisms. Stimulatory and inhibitory effects are represented by pointed black and blunt grey arrows, respectively. APOBEC editing of single strand miRNAs associated with AGO can affect miRNA activity or change miRNA specificity to different target mRNAs (left side of the panel). On the other hand, APOBEC activity can also directly impact the RISC complex and perturb its assembly (right side of the panel). (**c**) APOBEC/ADAR-mediated miRNA editing can modulate miRNAs activity through different mechanisms. First, editing of the seed sequence on miRNAs can change their specificities for given mRNAs or ceRNAs (upper panels). Second, editing of the MRE on mRNA targets can either impede the recognition and binding of miRNAs (middle panels), or on the contrary, modify MREs, thereby allowing binding of miRNAs that were not recognizing the unedited MRE (lower panels).

**Figure 4 ijms-20-06249-f004:**
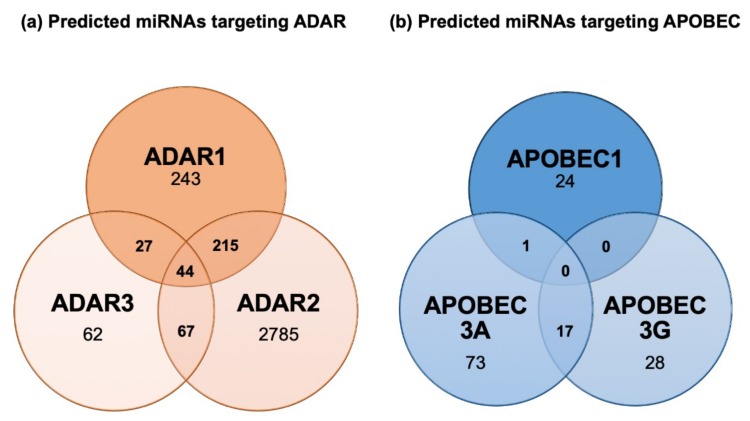
Predicted human miRNAs potentially regulating *ADARs* and *APOBECs* expressions. (**a**) Predicted human miRNAs targeting the three *ADAR* isoforms. (**b**) Predicted human miRNAs targeting the three *APOBEC* isoforms. The lists of predicted miRNAs modulating *ADARs* or *APOBECs* were retrieved from the miRWalk 2.0 database. Predicted miRNAs were obtained using 12 different algorithms (i.e., miRWalk2.0, MicroT4, miRanda, miRBridge, miRDB, miRMap, miRNAMap, PICTAR2, PITA, RNA22, RNAhybrid and TargetScan). Only miRNAs predicted by at least five different algorithms were considered.

**Figure 5 ijms-20-06249-f005:**
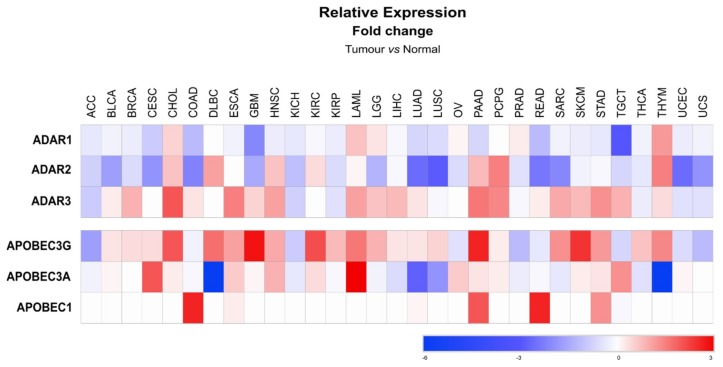
Expression patterns of *ADAR* and *APOBEC* editing enzymes in tumors. The heat map displays the mean fold change, following a log2 transformation, of *ADAR* and *APOBEC* gene expression in tumors from different tissues compared to their related non-tumoral tissues. Expression levels of each gene were retrieved using GEPIA2 database (http://gepia2.cancer-pku.cn/#index), which includes non-tumoral tissue and tumor samples from the Cancer Genome Atlas (TCGA). Fold changes were calculated using the ratio of tumor versus non-tumoral expression levels. Heat map was designed using Morpheus (https://software.broadinstitute.org/morpheus). Tumors are identified according with the TCGA nomenclature: ACC—adrenocortical carcinoma; BLCA—bladder urothelial carcinoma; BRCA—breast invasive carcinoma; CESC—cervical squamous cell carcinoma and endocervical adenocarcinoma; CHOL—cholangiocarcinoma; COAD—colon adenocarcinoma; DLBC—lymphoid neoplasm diffuse large B-cell lymphoma; ESCA—esophageal carcinoma; GBM—glioblastoma multiform; HNSC—head and neck squamous cell carcinoma; KICH—kidney chromophobe; KIRC—kidney renal clear cell carcinoma; KIRP—kidney renal papillary cell carcinoma; LAML – lymphoblastic acute myeloid leukemia; LGG—brain lower grade glioma; LIHC—liver hepatocellular carcinoma; LUAD—lung adenocarcinoma; LUSC—lung squamous cell carcinoma; MESO—mesothelioma; OV—ovarian serous cystadenocarcinoma; PAAD—pancreatic adenocarcinoma; PCPG—pheochromocytoma and paraganglioma; PRAD—prostate adenocarcinoma; READ—rectum adenocarcinoma; SARC—sarcoma; SKCM—skin cutaneous Melanoma; STAD—stomach adenocarcinoma; TGCT—testicular germ cell tumors; THCA—thyroid carcinoma; THYM—thymoma; UCEC—uterine corpus endometrial carcinoma; UCS—uterine carcinosarcoma; UVM—uveal melanoma.

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
