# Peer review of "Deciphering miRNAs’ Action through miRNA Editing"

_ijms, 2019, doi:10.3390/ijms20246249_

Round 1

Reviewer 1 Report

The paper by Correia de Sousa and coworkers is a review describing the current knowledge about the functional implications of RNA-editing mechanisms over miRNA function. The review is excellent, with a clear organization and elegant figures. I can state that this is a great example of a paper easy to read, well organized and with extremely relevant contents that are not frequently available. For this reason, I will recommend its publication in IJMS.

I have a very minor comments/questions for the authors to address before publication. Congratulations for such a great piece of work.

Minor comments:

1.- When the authors discuss the promiscuity of the miRNA action by targeting multiple mRNAs, they talk about the location of the target mRNAs in different cell compartments (Lines 92-94). Can the authors discuss a little bit more about, citing some references if possible?

2.- I would recommend the authors to include a paragraph or two with a deeper critical discussion about the threads and advantages of current available methods to infer the RNA editing (either in silico or wet-lab based methods), when applied to the miRNA field. It would be very interesting to have the critical and expert opinion of the authors about the application of different methods. I was wondering if already existing NGS data (small RNA-seq) could be analyzed to determine the RNA editing patterns. The authors commented about the difficulties of applying this method because of the small size of miRNAs and difficulties of isolation, but these facts are easily overcame with the current methods for small-RNA library preparation before sequencing. Is there any further detail to be considered?. Please discuss with more detail.

Author Response

Name of Journal: International Journal of Molecular Sciences

Manuscript Type: Review

Manuscript ID: ijms-644282

Manuscript title: Deciphering miRNAs action through miRNA editing

Correspondence to: Pr. Michelangelo Foti, Department of Cell Physiology and Metabolism, Faculty of Medicine, University of Geneva, CMU, 1 rue Michel-Servet, 1211 Geneva, Switzerland. Tel.: +41 22 3795204; fax: +41 22 3795260. E-mail address: Michelangelo.foti@unige.ch.

Dear Dr. Peng, dear editors,

We would like to thank you and the reviewers for the positive and constructive comments that helped us improve the quality of our manuscript. All comments and suggestions of the reviewers have been addressed and the manuscript has been improved accordingly. We hope that both you and the reviewers will be satisfied with our revisions and that our manuscript is now appropriated for publication in IJMS. Comments of the reviewers are addressed below and revisions are highlighted in red in new version of our manuscript.

Reviewer 1: Comments and Suggestions for Authors

The paper by Correia de Sousa and coworkers is a review describing the current knowledge about the functional implications of RNA-editing mechanisms over miRNA function. The review is excellent, with a clear organization and elegant figures. I can state that this is a great example of a paper easy to read, well organized and with extremely relevant contents that are not frequently available. For this reason, I will recommend its publication in IJMS.

I have a very minor comments/questions for the authors to address before publication. Congratulations for such a great piece of work.

Response to reviewer 1:

We would like to kindly thank the reviewer for her/his very positive comments and the constructive remarks. We have addressed the comments as following.

Minor comments:

1.- When the authors discuss the promiscuity of the miRNA action by targeting multiple mRNAs, they talk about the location of the target mRNAs in different cell compartments (Lines 92-94). Can the authors discuss a little bit more about, citing some references if possible?

The discussion about miRNAs compartmentalization has been implemented and 3 references have been added as suggested by the reviewer (lines 94 to 98).

2.- I would recommend the authors to include a paragraph or two with a deeper critical discussion about the threads and advantages of current available methods to infer the RNA editing (either in silico or wet-lab based methods), when applied to the miRNA field. It would be very interesting to have the critical and expert opinion of the authors about the application of different methods. I was wondering if already existing NGS data (small RNA-seq) could be analyzed to determine the RNA editing patterns. The authors commented about the difficulties of applying this method because of the small size of miRNAs and difficulties of isolation, but these facts are easily overcame with the current methods for small-RNA library preparation before sequencing. Is there any further detail to be considered?. Please discuss with more detail.

As suggested by the reviewers, we have now extended our discussions about methodologies (in silico or wet-lab based methods) currently used in the field by adding several paragraphs addressing the points of the reviewer and new references in the chapter 5 “Tools to study miRNA editing” of our review (lines 300 to 363). We hope that the reviewer will find the new added information useful for the reader.

Reviewer 2 Report

General Comments

The manuscript by Correira de Sousa et al is discussing very interesting and important topic. Important because is every day clearer the importance and the relevance of RNA editing in both physiological and pathological processes. Interesting, because they discuss in detail the possible roles of RNA editing in miRNA processing and function. I want to congratulate the authors because they did a very good job, the article is well written, clear and well organized.

Below few specific suggestions to improve further this well-done work.

Specific Comments

Line 133:

“RNA (dsRNA) is mostly catalyzed by ADAR”

If the authors are talking about mRNA, ADAR is the only family inducing A-to-I editing (I would avoid mostly). If the authors means also tRNA, they need to mention also ADAT family.

Line 136-137:

“Both ADAR1 and ADAR2 genes are ubiquitously expressed to different extent depending on the tissue or cell type”.

As far as I know ADAR1 is ubiquitous, but ADAR2 is not (I would say that is mainly expressed in the nervous system). Maybe the authors want to double check in the literature about this point. 

Line 139:

“the full length ADAR1p150 accumulates in the nucleus but can translocate in the cytoplasm”

It is true that p150 is able to shuttle between nucleus and cytoplasm (as p110) but thus isoform is mainly cytoplasmic, and maybe the authors want to mention that p150 is also IFN dependent. For more information and REFs see “Xu and Öhman 2018” at page 2.

Line 143-144:

“ADAR1 and ADAR2 are not viable and die from neurological disorders during embryonic development or shortly after birth”.

This is true for ADAR2, but ADAR1 ko mice die due to strong activation of innate immune response (see Pestal et al 2015).

Line 297-298:

“i.e. CRISPR-Cas13 or RNA Editing for Programmable A-to-I Replacement (REPAIR) system”

This is only the most recent tool developed. There are at least other 2 technologies capable to do the same, lambda-N-ADAR2 (see publications from Rosenthal’s lab – eg Montiel-Gonzalez et al 2013) and SNAP-ADAR (Stafforst’s lab). Considering that these technologies have been developed previously the REPAIR system, the authors must mention them.

Author Response

(The authors gave the same response as above.)
